# Fair Multiple Decision Making Through Soft Interventions

**Yaowei Hu**
University of Arkansas
yaoweihu@uark.edu

**Yongkai Wu**
Clemson University
yongkaw@clemson.edu

**Lu Zhang**
University of Arkansas
lz006@uark.edu

**Xintao Wu**
University of Arkansas
xintaowu@uark.edu

## Abstract

Previous research in fair classification mostly focuses on a single decision model. In reality, there usually exist multiple decision models within a system and all of which may contain a certain amount of discrimination. Such realistic scenarios introduce new challenges to fair classification: since discrimination may be transmitted from upstream models to downstream models, building decision models separately without taking upstream models into consideration cannot guarantee to achieve fairness. In this paper, we propose an approach that learns multiple classifiers and achieves fairness for all of them simultaneously, by treating each decision model as a soft intervention and inferring the post-intervention distributions to formulate the loss function as well as the fairness constraints. We adopt surrogate functions to smooth the loss function and constraints, and theoretically show that the excess risk of the proposed loss function can be bounded in a form that is the same as that for traditional surrogated loss functions. Experiments using both synthetic and real-world datasets show the effectiveness of our approach.

## 1 Introduction

How to ensure fairness in algorithmic decision making models is an important task in machine learning [12, 15]. Over the past years, many researchers have been devoted to the design of fair classification algorithms with respect to a pre-defined protected attribute, such as race or sex, and a decision task/model, such as hiring [1, 11, 24]. In particular, one line of the work is to incorporate fairness constraints into classic learning algorithms to build fair classifiers from potentially biased data [4, 13, 29, 31–33]. Most of previous research generally focuses on a single decision model. However, in reality there usually exist multiple decision models within a system and all of which may contain a certain amount of discrimination, either introduced by themselves or transmitted from upstream models. As a motivating example, consider two decision tasks $Y_1, Y_2$ where $Y_1$ is used by the city government to allocate policing resources to different locations and $Y_2$ is used by a local bank to make personal loan decisions. Due to historically segregated housing, neighborhood racial composition differs based on geographic locations, and there can exist direct racial discrimination in $Y_1$ as well. Thus, certain locations will be allocated more police resources than others, resulting in larger numbers of criminal arrest records. As a result, when the criminal arrest record is used in $Y_2$, certain racial group will receive unfair disadvantage in getting loans.

Ideally we would like to build fair models for all decision making tasks. However, if decision models influence one another, it is not a straightforward problem even if we know how to build a fair model for each task. This is because the data distribution can change as a consequence of deploying

new models. If we build the model for each task independently using static training datasets, the learning process of each model is based on the fixed distribution given in the training data. However, deploying new fair models would change the distributions of attribute variables that are affected by their decisions as well as the discrimination that is passing down. As a result, the subsequent models built on the original distribution may not perform well in terms of both accuracy and fairness. On the other hand, if we build fair models one by one following a temporal sequential order, each time deploying a model and collecting the output data before building the next one, then the time needed for building all models may not be acceptable for some applications.

In this paper, we propose an approach that learns multiple fair classifiers simultaneously and only requires a static training dataset. The core idea is to leverage Pearl's structural causal model (SCM) [23], treat each decision model as a soft intervention and infer the post-intervention distributions to formulate the loss function as well as the fairness constraints. The SCM is widely adopted in fair classification research for defining fairness as the causal effect of the protected attribute on the decision [17, 21, 28, 30, 34–36]. Causal inference in the SCM is often facilitated with the "(hard) intervention" that forces some variable $X$ to take certain constant $x$, denoted by $do(X = x)$ [23]. "Soft intervention" [6, 16], also known as the "conditional action" [23], extends the hard intervention such that variable $X$ is forced to take a specified functional relationship $g(z)$ in responding to some set $Z$ of other variables, denoted by $do(X = g(z))$. In our approach, the deploying of new decision models is considered as to perform soft interventions on the decisions, whose influence can be inferred as the post-intervention distributions. By quantifying fairness as causal effects of the protected attribute on all decisions, under the hard intervention on the protected attribute and soft interventions on decisions, we formulate fair classification for multiple decisions as a single constrained optimization problem.

Combining multiple decision models together makes the optimization challenging to solve. Similarly to [29], we adopt surrogate functions to smooth the loss function and constraints. However, the difference in our problem is that, each decision model is associated with a surrogate function, and the surrogated protections are used in downstream decision models, resulting non-linear combinations of multiple surrogate functions. As a result, our loss function is different from traditional surrogated loss functions whose excess risks have been analyzed and bounded in [2]. To investigate the excess risk of our loss function, we adopt theoretical tools in [2] and show that nontrivial upper bounds exist on the excess risk in a form that is the same as that for traditional surrogated loss functions given in [2], irrespective of the number of decision models involved.

**Related work.** How to ensure fairness in a compound decision-making process, called pipeline or multi-stage selection, has received attention in [3, 8–10]. Pipelines differ from our multi-decision setting in that individuals drop out at any stage and classification in subsequent stages depends on the remaining cohort of individuals. In [27], the authors assume that multiple functions over the same set of attributes are multiplied to produce an overall score. Other related but different works include long term fairness (e.g., [20]), which concerns for how decisions affect the long-term well-being of disadvantaged groups measured in terms of a temporal variable of interest, and fair sequential learning (e.g., [14]), which sequentially considers each individual and makes decision for them.

**Contributions.** To the best of our knowledge, this is the first work to study fair multiple decision making where the feature distribution may change due to the deployment of decision models. Our approach provides a general way to incorporate fairness constraints into the generic classification formulation such that we can readily employ off-the-shelf classification models and optimization algorithms. The causal inference allows us to train all decision models jointly from a single dataset. Since our approach is based on the SCM, all SCM-based fairness notions, including the total effect [35], direct and indirect discrimination [21, 35, 36], counterfactual fairness [17, 28, 34], and PC-fairness [30], can be naturally applied to our problem formulation. The theoretical results imply that we don't need to worry about additional losses caused by multiple surrogate functions. By conducting experiments on both synthetic and real-world datasets, we show that our approach consistently outperforms the approach which builds fair classifiers for each decision separately.

## 2   Preliminaries

We use an upper case letter, e.g., $X$, to denote a variable, and use a lower case letter, e.g., $x$, to denote a value of $X$. We use bold letters, e.g., $\mathbf{X}$ and $\mathbf{x}$, to denote a set of variables and their values.

## 2.1 Structural Causal Model

A structural causal model (SCM) is formally defined by a triple $\mathcal{M} = \langle \mathbf{U}, \mathbf{V}, \mathbf{F} \rangle$ where $\mathbf{U}$ is a set of exogenous variables, $\mathbf{V}$ is a set of endogenous variables, and $\mathbf{F}$ is a set of structural equations mapping $\mathbf{V} \times \mathbf{U} \mapsto \mathbf{V}$. Specifically, for each $V \in \mathbf{V}$, there is an equation $V = f_V(\mathsf{pa}_V, u_V)$ where $\mathsf{pa}_V$ is a realization of a set of endogenous variables $\mathsf{PA}_V \subseteq \mathbf{V} \backslash V$ called the parents of $V$. If all exogenous variables in $\mathbf{U}$ are assumed to be mutually independent (i.e., no hidden confounder), then the SCM is called a Markovian model. In this paper, we assume that the SCMs we are dealing with are Markovian models. The SCM $\mathcal{M}$ can be illustrated by a causal graph $\mathcal{G} = \langle \mathbf{V}, \mathbf{E} \rangle$ where each node in $\mathbf{V}$ represents an endogenous variable and each arc in $\mathbf{E}$, denoted by an arrow $\rightarrow$ pointing from one node to another, represents the parental relationship defined in structural equations. Each node $V$ is associated with a conditional distribution given its parents, i.e., $P(v|\mathsf{pa}_V)$ to reflect the relationship defined by equation $f_V$.

The hard intervention that forces variable $V$ to take constant $v$, denoted by $do(V = v)$ or $do(v)$ for short, is performed by substituting equation $V = f_V(\mathsf{pa}_V, u_V)$ with $V = v$. The post-intervention distribution of a variable $W$ other than $V$ is denoted by $P(w|do(v))$. The soft intervention extends the hard intervention such that it forces variable $V$ to take functional relationship $g(\mathbf{z})$ in responding to some other variables $\mathbf{Z}$, which can be similarly denoted by $do(V = g(\mathbf{z}))$. In other words, it substitutes equation $V = f_V(\mathsf{pa}_V, u_V)$ with a new function $V = g(\mathbf{z})$. The difference between hard intervention $do(V = v)$ and soft intervention $V = g(\mathbf{z})$ is that, after the hard intervention $V$ becomes a constant (or be associated with a distribution of $P(V = v) = 1$), but after the soft intervention $V$ can be associated with an arbitrary distribution $P_g(v|\mathbf{z})$ determined by function $g$. In particular, depending on the intervention, function $g$ could receive as inputs the variables $\mathbf{Z}$ other than the original parents $\mathsf{pa}_V$, as long as they are not the descendants of $V$.

## 2.2 Fair Classification

Following the notations used in [29], the problem of fair classification is to learn a mapping $f : \mathbf{X} \mapsto Y$ parameterized with $\theta$, where $\mathbf{X}$ is a set of input attributes and $Y = \{0, 1\}$ is the class label. The learning algorithm aims to minimize the classification error $\mathbb{E}_{\mathbf{X},Y}[\mathbb{1}_{f(\mathbf{x}) \neq y}]$, where $\mathbb{1}_A$ is the indicator function, i.e., $\mathbb{1}_A = 1$ if $A$ is true and $\mathbb{1}_A = 0$ if $A$ is false. Usually, $f$ is defined based on another function $h$ that is performed in the real number domain, i.e., $h : \mathbf{X} \mapsto \mathbb{R}$ and $f(\mathbf{x}) = \mathbb{1}_{h(\mathbf{x}) \geq 0}$. Thus, the classification error can be reformulated as

$$R(h) = \mathbb{E}_{\mathbf{X}} \left[ P(Y = 1|\mathbf{x}) \mathbb{1}_{h(\mathbf{x}) < 0} + P(Y = 0|\mathbf{x}) \mathbb{1}_{h(\mathbf{x}) \geq 0} \right]. \tag{1}$$

By using surrogate functions $\phi(\cdot)$ to smooth and bound the indicator function (i.e., the 0-1 loss), we obtain the $\phi$-loss as:

$$R_\phi(h) = \mathbb{E}_{\mathbf{X}} \left[ P(Y = 1|\mathbf{x}) \phi(h(\mathbf{x})) + (1 - P(Y = 1|\mathbf{x})) \phi(-h(\mathbf{x})) \right],$$

and the optimization problem as $\min_{h \in \mathcal{H}} R_\phi(h)$. Similarly fair classification can be formulated as a constrained optimization problem

$$\min_{h \in \mathcal{H}} \quad R_\phi(h) \quad \text{s.t.} \quad -\tau \leq T_\phi(h) \leq \tau, \tag{2}$$

where $T_\phi(h)$ is a measure of $\phi$-unfairness depending on the particular fairness notion used which is also smoothed by using surrogate functions. Widely used surrogate functions include the hinges loss, square loss, logistic loss, exponential loss, etc.

## 3 Formulating Fair Classification for Making Multiple Decisions

In this section, we formally formulate the fair classification problem for making multiple decisions. Consider a protected attribute $S$, a set of non-protected attributes $\mathbf{X} = \{X_1, \cdots, X_m\}$ and a set of decisions $\mathbf{Y} = \{Y_1, \cdots, Y_l\}$. For ease of representation, we assume that the protected attribute and all decisions are binary, i.e., $S = \{s^-, s^+\}$ with $s^-$ denoting the protected group and $s^+$ denoting the non-protected group, and $Y_k = \{y^-, y^+\}$ for each $Y_k \in \mathbf{Y}$ with $y^-$ denoting the negative decision (i.e., $Y_k = 0$) and $y^+$ denoting the positive decision (i.e., $Y_k = 1$). Often we abbreviate expressions $Y_k = y^-, y^+$ as $y_k^-, y_k^+$. Note that decisions can be interdependent such that later decisions may depend on the consequences of earlier decisions either directly and/or indirectly through the change

of some features that is mediated between the two decisions. In real situations, such indirect influence may need time to take effect and cannot be observed within a short period of time. Therefore, we only assume that a historical dataset $\mathcal{D} = \{(s^{(i)}, \mathbf{x}^{(i)}, \mathbf{y}^{(i)})\}_{i=1}^{N}$ that reflects the original decision making mechanisms is observed.

Our task is to build a classifier $h_k$ for each decision $Y_k$ from training data $\mathcal{D}$. Classifier $h_k$ takes some profile attributes $\mathbf{Z}_k \subseteq \{S\} \cup \mathbf{X}$ as the input to make the prediction as $\mathbb{1}_{h_k(\mathbf{z}_k) \geq 0}$. We would like to ensure that any classifier is fair if all classifiers are deployed, with the fairness of all classifiers being measured using the same fairness notion. In this paper, for simplicity we consider total effect [35] as the fairness notion which is defined based on the total causal effect as follows. Nevertheless, our formulation can be easily extended to other causal-based fairness notions as long as they can be identified and computed with expressions of observational distributions.

**Definition 1.** *For the classifier built for each decision $Y_k$, it is considered to be fair if*
$$-\tau \leq P^*(y_k^+|do(s^+)) - P^*(y_k^+|do(s^-)) \leq \tau$$
*where $\tau$ is a user-defined threshold and $P^*$ is the distribution after all classifiers are deployed.*

As shown in [29], the formulation of fair classification consists of a loss function for quantifying the classification error and a number of constraints for enforcing fairness. For the case of a single decision model, the loss function can be directly computed from $\mathcal{D}$, and fairness constraints can be computed from $\mathcal{D}$ as well after performing hard intervention $do(s)$. However, for the case of multiple decision models, due to the change in the data distribution made by model deployment, the loss function and fairness constraints should not be computed from $\mathcal{D}$ but $P^*$ which may be different from the distribution followed by $\mathcal{D}$. Therefore, we propose to adopt the soft intervention to model all model deployments and infer post-intervention distributions.

To this end, we build a causal graph $\mathcal{G}$ to represent the causal structure of the underlying data generation mechanism from dataset $\mathcal{D}$. The research of causal structure discovery is quite active in recent years and many algorithms have been proposed [26]. Given the causal graph, we capture the deployment of classifier $h_k(\mathbf{z}_k)$ as a soft intervention that forces the prediction of decision $Y_k$ to take functional relationship $h_k(\mathbf{z}_k)$, denoted as $do(h_k)$. Consequently, distribution $P^*$ after the deployment of all classifiers can be captured by the post-intervention distribution after performing soft interventions $do(h_1, \cdots, h_l)$. Then, the classification error of $h_k(\mathbf{z}_k)$ after the deployment of classifiers could be measured similarly to Eq. (1) as given by
$$R(h_k) = \mathop{\mathbb{E}}_{\mathbf{Z}_k|do(h_1,\cdots,h_l)} \left[ P(y_k^+|\mathbf{z}_k)\mathbb{1}_{h_k(\mathbf{z}_k)<0} + P(y_k^-|\mathbf{z}_k)\mathbb{1}_{h_k(\mathbf{z}_k)\geq 0} \right], \tag{3}$$
where the expectation is computed on the post-intervention distribution of $\mathbf{Z}_k$. Similarly, the fairness constraints of $h_k(\mathbf{z}_k)$ is given by the total effect
$$T(h_k) = P(y_k^+|do(s^+, h_1, \cdots, h_l)) - P(y_k^+|do(s^-, h_1, \cdots, h_l)), \tag{4}$$
which is based on the post-intervention distributions of $Y_k$ after performing both hard intervention $do(s)$ and soft interventions $do(h_1, \cdots, h_l)$. [1]

Take the toy model given in the introduction as an example, where there are two decisions $Y_1$ and $Y_2$ representing the policing resource allocation and bank loan decision respectively. We treat race as the protected attribute, denoted by $S$. We denote the location of the residential area as $X_1$, and denote the number of criminal arrest records in each area as $X_2$. The causal graph of this toy model is shown in Figure 1. We would like to build two fair classifiers

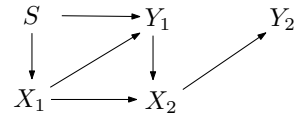

Figure 1: Causal graph of toy model.

$h_1(x_1)$ and $h_2(x_2)$ for predicting $Y_1$ and $Y_2$. Note that the inputs of classifiers could be different from the original parents of $Y_1$ and $Y_2$, which are $\{S, X_1\}$ and $X_2$ respectively. Their loss functions are given by $R(h_1)$, $R(h_2)$, and fair constraints are given by $T(h_1)$, $T(h_2)$.

Next, we need to derive $R(h_k)$ and $T(h_k)$, which are given on the post-intervention distribution, as smooth expressions on $\mathcal{D}$, which is the observational data. By using surrogate functions $\phi(\cdot)$ to smooth and bound the indicator function, we finally derive the formulas of $R_\phi(h_k)$ and $T_\phi(h_k)$ that will be used in our formulation of fair multi-decision learning.

## 3.1 Deriving Loss Function and Fair Constraints

In the above example, the learning of classifier $h_1$ could be done by solving an ordinary fair classification problem. However, when learning classifier $h_2$, both its loss and fairness are affected by classifier $h_1$, and in this case the effect is transmitted indirectly through $X_2$. To accurately measure the loss and fairness of $h_2$, we need to mathematically express the effect of $h_1$ as post-intervention distributions. Thus, we apply following three properties of the (soft) intervention to compute post-intervention distributions from observational data.

    (1) An intervention on a variable $V$ would not change the distribution of $V$'s non-descendant.
    (2) An intervention on $V$ would not change the generation mechanism of another variable $W$, i.e., distribution $P(w|\mathsf{pa}_W)$ would not be changed.
    (3) A soft intervention on $V$ would change its conditional distribution $P(v|\mathsf{pa}_V)$ according to the defined functional relationship.

Next we show how the properties work in the toy example. Note that $R(h_2)$ is given by $\mathbb{E}_{X_2|do(h_1,h_2)}\left[P(y_2^+|x_2)\mathbb{1}_{h_2(x_2)<0} + P(y_2^-|x_2)\mathbb{1}_{h_2(x_2)\geq0}\right]$, which by definition is equal to $\sum_{X_2} P(x_2|do(h_1,h_2))\left[P(y_2^+|x_2)\mathbb{1}_{h_2(x_2)<0} + P(y_2^-|x_2)\mathbb{1}_{h_2(x_2)\geq0}\right]$. Due to Property (1), $P(x_2|do(h_1,h_2)) = P(x_2|do(h_1))$, which can be broken down by conditioning on $X_1, Y_1$ as $\sum_{X_1,Y_1} P(x_2|do(h_1),x_1,y_1)P(x_1,y_1|do(h_1))$. Due to Property (2), we have that $P(x_2|do(h_1),x_1,y_1) = P(x_2|x_1,y_1)$. Meanwhile, we rewrite $P(x_1,y_1|do(h_1))$ as $P(x_1|do(h_1))P(y_1|do(h_1),x_1)$ which is equal to $P(x_1)P(y_1|do(h_1),x_1)$. Then, we further break down $P(y_1|do(h_1),x_1)$ as $\sum_S P(y_1|do(h_1),s,x_1)P(s|do(h_1),x_1)$. Due to Property (1), we have $P(s|do(h_1),x_1) = P(s|x_1)$. Due to Property (3), we have $P(y_1|do(h_1),s,x_1)$ be equal to a new distribution $P_{h_1}(y_1|x_1)$ defined by function $h_1$, which is given by $\mathbb{1}_{h_1(x_1)\geq0}$ if $y_1 = y_1^+$ and $\mathbb{1}_{h_1(x_1)<0}$ if $y_1 = y_1^-$ in our case. Finally, combining every components above together and using a surrogate function $\phi$ to replace each indicator, we obtain that $R_\phi(h_2) =$

$$\sum_{S,X_1,X_2} P(s,x_1)\big(\phi(h_2(x_2))\phi(-h_1(x_1))P(y_2^+|x_2)P(x_2|x_1,y_1^+) + \phi(h_2(x_2))\phi(h_1(x_1))P(y_2^+|x_2)P(x_2|x_1,y_1^-)$$
$$+\phi(-h_2(x_2))\phi(-h_1(x_1))P(y_2^-|x_2)P(x_2|x_1,y_1^+) + \phi(-h_2(x_2))\phi(h_1(x_1))P(y_2^-|x_2)P(x_2|x_1,y_1^-)\big).$$

For $T(h_2)$, it is given by $P(y_1^+|do(s^+,h_1,h_2)) - P(y_1^+|do(s^-,h_1,h_2))$, which can be directly rewritten as $P(y_1^+|do(s^+,h_1)) + P(y_1^-|do(s^-,h_1)) - 1$. By similarly applying the three properties, we could obtain that $T_\phi(h_2) =$

$$\sum_{X_1,X_2} \big(\phi(-h_2(x_2))\phi(-h_1(x_1))P(x_1|s^+)P(x_2|x_1,y_1^+) + \phi(-h_2(x_2))\phi(h_1(x_1))P(x_1|s^+)P(x_2|x_1,y_1^-)$$
$$+\phi(h_2(x_2))\phi(-h_1(x_1))P(x_1|s^-)P(x_2|x_1,y_1^+) + \phi(h_2(x_2))\phi(h_1(x_1))P(x_1|s^-)P(x_2|x_1,y_1^-)\big) - 1.$$

More generally, when there are $l$ classifiers, we could derive $R_\phi(h_k)$ and $T_\phi(h_k)$ by using the factorization formula proposed in [16] which implicitly encode all three properties. For $R_\phi(h_k)$, according to the factorization formula, post-intervention $P(\mathbf{z}_k|do(h_1),\cdots,do(h_l))$ is given by[2]

$$P(\mathbf{z}_k|do(h_1,\cdots,h_l)) = \sum_{\mathbf{X}\backslash\mathbf{Z}_k,\mathbf{Y}} \prod_{i=1}^{l} P_{h_i}(y_i|\mathbf{z}_i) \prod_{i=1}^{m} P(x_i|\mathsf{pa}_{X_i}), \tag{5}$$

where $P_{h_i}(y_i|\mathbf{z}_i)$ is the distribution of $Y_i$ defined by classifier $h_i(\mathbf{z}_i)$, i.e., $\mathbb{1}_{h_i(\mathbf{z}_i)\geq0}$ if $y_i = y^+$ and $\mathbb{1}_{h_i(\mathbf{z}_i)<0}$ if $y_i = y^-$. Note that all terms in Eq. (5) can be computed from data. Then, we can derive the formula for computing $R_\phi(h_k)$.

However, it may not be ideal to directly apply Eq. (5) to our problem formulation. First, some computations in Eq. (5) are not necessary since $\mathbf{Z}_k$ should not be affected by interventions on the non-ancestors of $Y_k$. More importantly, if any $X_i$ is a continuous variable, its corresponding summation in Eq. (5) would become an integral, making the gradient difficult to compute. Thus, to further simplify Eq. (5), we index all attributes in $\mathbf{X}$ and $\mathbf{Y}$ according to the topological ordering, and denote the

subsets of $\mathbf{X}$ and $\mathbf{Y}$ that are prior to $Y_i$ (or $X_i$) in the topological order as $\mathbf{X}'_{Y_i}$ and $\mathbf{Y}'_{Y_i}$ ($\mathbf{X}'_{X_i}$ and $\mathbf{Y}'_{X_i}$). Then, by canceling out all terms that are after $Y_k$ in the order, it follows that

$$
\begin{aligned}
P(\mathbf{z}_k|do(h_1,\cdots,h_l)) &= \sum_{\{S,\mathbf{X}'_{Y_k}\}\backslash\mathbf{Z}_k,\mathbf{Y}'_{Y_k}} P(s) \prod_{Y_i\in\mathbf{Y}'_{Y_k}} P_{h_i}(y_i|\mathbf{z}_i) \prod_{X_i\in\mathbf{X}'_{Y_k}} P(x_i|\mathsf{pa}_{X_i}) \\
&= \sum_{\{S,\mathbf{X}'_{Y_k}\}\backslash\mathbf{Z}_k,\mathbf{Y}'_{Y_k}} P(s) \prod_{Y_i\in\mathbf{Y}'_{Y_k},y_i^+} \mathbb{1}_{h_i(\mathbf{z}_i)\geq 0} \prod_{Y_i\in\mathbf{Y}'_{Y_k},y_i^-} \mathbb{1}_{h_i(\mathbf{z}_i)<0} \prod_{X_i\in\mathbf{X}'_{Y_k}} P(x_i|\mathsf{pa}_{X_i}).
\end{aligned}
\tag{6}
$$

We can rewrite $P(s)\prod_{X_i\in\mathbf{X}'_{Y_k}} P(x_i|\mathsf{pa}_{X_i})$ as

$$
P(s)\prod_{X_i\in\mathbf{X}'_{Y_k}} P(x_i|s,\mathbf{x}'_{X_i},\mathbf{y}'_{X_i}) = P(s)\prod_{X_i\in\mathbf{X}'_{Y_k}} P(x_i|s,\mathbf{x}'_{X_i})\frac{P(\mathbf{y}'_{X_i}|s,x_i,\mathbf{x}'_{X_i})}{P(\mathbf{y}'_{X_i}|s,\mathbf{x}'_{X_i})} = P(s,\mathbf{x}'_{X_i})\prod_{X_i\in\mathbf{X}'_{Y_k}} \frac{P(\mathbf{y}'_{X_i}|s,x_i,\mathbf{x}'_{X_i})}{P(\mathbf{y}'_{X_i}|s,\mathbf{x}'_{X_i})}.
$$

Thus, we can rewrite Eq. (6) as an expectation over $S, \mathbf{X}'_{Y_k}$. With the surrogate function we obtain

$$
\begin{aligned}
R_\phi(h_k) = \mathop{\mathbb{E}}_{S,\mathbf{X}'_{Y_k}} \Bigg[ &P(y_k^+|\mathbf{z}_k)\phi(h_k(\mathbf{z}_k)) \sum_{\mathbf{Y}'_{Y_k}} \prod_{Y_i\in\mathbf{Y}'_{Y_k},y_i^+} \phi(-h_i(\mathbf{z}_i)) \prod_{Y_i\in\mathbf{Y}'_{Y_k},y_i^-} \phi(h_i(\mathbf{z}_i)) \prod_{X_i\in\mathbf{X}'_{Y_k}} \frac{P(\mathbf{y}'_{X_i}|s,x_i,\mathbf{x}'_{X_i})}{P(\mathbf{y}'_{X_i}|s,\mathbf{x}'_{X_i})} \\
&+ P(y_k^-|\mathbf{z}_k)\phi(-h_k(\mathbf{z}_k)) \sum_{\mathbf{Y}'_{Y_k}} \prod_{Y_i\in\mathbf{Y}'_{Y_k},y_i^+} \phi(-h_i(\mathbf{z}_i)) \prod_{Y_i\in\mathbf{Y}'_{Y_k},y_i^-} \phi(h_i(\mathbf{z}_i)) \prod_{X_i\in\mathbf{X}'_{Y_k}} \frac{P(\mathbf{y}'_{X_i}|s,x_i,\mathbf{x}'_{X_i})}{P(\mathbf{y}'_{X_i}|s,\mathbf{x}'_{X_i})} \Bigg].
\end{aligned}
\tag{7}
$$

We can see that, in Eq. (7), only probabilities of categorical decisions are involved, and the expectation can be estimated as an empirical risk.

Similarly, for $T(h_k)$, Eq. (4) can be directly rewritten as $T(h_k) = P(y_k^+|do(s^+,h_1,\cdots,h_l)) + P(y_k^-|do(s^-,h_1,\cdots,h_l)) - 1$, and $P(y_k^+|do(s,h_1,\cdots,h_l))$ can be given by

$$
\mathop{\mathbb{E}}_{\mathbf{X}'_{Y_k}|S=s} \Bigg[ \mathbb{1}_{h_k(\mathbf{z}_k)>0} \sum_{\mathbf{Y}'_{Y_k}} \prod_{Y_i\in\mathbf{Y}'_{Y_k},y_i^+} \mathbb{1}_{h_i(\mathbf{z}_i)>0} \prod_{Y_i\in\mathbf{Y}'_{Y_k},y_i^-} \mathbb{1}_{h_i(\mathbf{z}_i)<0} \prod_{X_i\in\mathbf{X}'_{Y_k}} \frac{P(\mathbf{y}'_{X_i}|s,x_i,\mathbf{x}'_{X_i})}{P(\mathbf{y}'_{X_i}|s,\mathbf{x}'_{X_i})} \Bigg].
$$

By applying surrogate function $\phi$, we obtain that

$$
\begin{aligned}
T_\phi(h_k) = &\mathop{\mathbb{E}}_{\mathbf{X}'_{Y_k}|S=s^+} \Bigg[ \phi(-h_k(\mathbf{z}_k)) \sum_{\mathbf{Y}'_{Y_k}} \prod_{Y_i\in\mathbf{Y}'_{Y_k},y_i^+} \phi(-h_i(\mathbf{z}_i)) \prod_{Y_i\in\mathbf{Y}'_{Y_k},y_i^-} \phi(h_i(\mathbf{z}_i)) \prod_{X_i\in\mathbf{X}} \frac{P(\mathbf{y}'_{X_i}|s^+,x_i,\mathbf{x}'_{X_i})}{P(\mathbf{y}'_{X_i}|s^+,\mathbf{x}'_{X_i})} \Bigg] \\
+ &\mathop{\mathbb{E}}_{\mathbf{X}'_{Y_k}|S=s^-} \Bigg[ \phi(h_k(\mathbf{z}_k)) \sum_{\mathbf{Y}'_{Y_k}} \prod_{Y_i\in\mathbf{Y}'_{Y_k},y_i^+} \phi(-h_i(\mathbf{z}_i)) \prod_{Y_i\in\mathbf{Y}'_{Y_k},y_i^-} \phi(h_i(\mathbf{z}_i)) \prod_{X_i\in\mathbf{X}} \frac{P(\mathbf{y}'_{X_i}|s^-,x_i,\mathbf{x}'_{X_i})}{P(\mathbf{y}'_{X_i}|s^-,\mathbf{x}'_{X_i})} \Bigg] - 1.
\end{aligned}
\tag{8}
$$

## 3.2 Problem Formulation

Now we are ready to formulate the classification problem. For each classifier $h_k(\mathbf{z}_k)$, we derive its $\phi$-loss $R_\phi(h_k)$ and $\phi$-unfairness $T_\phi(h_k)$. Then, we minimize the summation of the $\phi$-loss over all classifiers. Meanwhile, given a fairness threshold $\tau_k$, we want the $\phi$-unfairness to be bounded within the range $[-\tau_k, \tau_k]$, so we require that $-\tau_k \leq T_\phi(h_k) \leq \tau_k$. Generally, large thresholds indicate loose fairness requirements and small ones indicate strict fairness requirements. However, due to the application of surrogate functions, $\tau_k$ may not be equal to threshold $\tau$ that is placed on the original fairness metric (e.g., 0.05 on the total effect used in the literature). In practice, we need to test different values of $\tau_k$ in order to find a good balance between fairness and accuracy.

**Problem Formulation 1.** *The problem of fair multiple decision making for* $\mathbf{Y} = \{Y_1, \cdots, Y_l\}$ *is formulated as the following constrained optimization problem:*

$$\min_{h_1,\cdots,h_l \in \mathcal{H}} \sum_{k=1}^{l} R_\phi(h_k) \quad \text{s.t.} \quad \forall k, \quad -\tau_k \leq T_\phi(h_k) \leq \tau_k.$$

*where the formulas of* $R_\phi(h_k)$ *and* $T_\phi(h_k)$ *are shown in Eqs.* (7) *and* (8), *respectively.*

From Section 3.1 we see that both the loss function and constraints in this formulation involve non-linear combinations of surrogate functions. Specifically, for each $Y_k$, surrogate functions of $Y_i$ that are ancestors of $Y_k$ are involved as multiplications. In essence, this is because the surrogated predictions of one classifier are used in computing the loss of downstream classifiers. As each surrogate function is involved as a single term in the multiplication in Eqs. (7) and (8), the gradients of $R_\phi(h_k)$ and $T_\phi(h_k)$ can be easily computed. However, it is important to know whether such "passing down" process would accumulate surrogate errors and affect the accuracy of classification. We analyze the risk bound of the optimization in the next subsection.

### 3.3 Excess Risk Bound

Our main result of the excess risk bound is on the unconstrained optimization of the loss function. We show that for each classifier $h_k$, $\phi$-loss $R_\phi(h_k)$ approaching its unconstrained optimum $R_\phi^*$ indicates that classification error $R(h_k)$ also approaching its unconstrained optimum $R^*$, no matter how many classifiers are involved in the formulation. Although this result does not directly give the risk bound to our constrained optimization problem, it can be easily extended to the constrained situation if we treat the constraints as penalty terms that are to be added to the loss function.

We first formally define $R^*$ and $R_\phi^*$, which are optimums of $R(h_k)$ and $R_\phi(h_k)$ over all possible classifiers. By replacing each $h_i(\mathbf{z}_i)$ in $R(h_k)$ and $R_\phi(h_k)$ with a real-valued variable $\alpha_i$, we define that $R^* = \inf_{\forall i, \alpha_i \in \mathbb{R}} R(h_k)$ and $R_\phi^* = \inf_{\forall i, \alpha_i \in \mathbb{R}} R_\phi(h_k)$. Then, we define the generic $\phi$-conditional risk $C_\phi^\eta(\alpha)$, optimal $\phi$-conditional risk $H_\phi(\eta)$, constrained optimal $\phi$-conditional risk $H_\phi^-(\eta)$, and $\psi$-transform. All these definitions are consistent to those in [2]. Let $\eta$ be a value in $[0, 1]$, $\bar{\eta} = 1 - \eta$, $\alpha$ be an arbitrary real value, and $\phi$ be a surrogate function, we define that $C_\phi^\eta(\alpha) = \eta\phi(\alpha) + \bar{\eta}\phi(-\alpha)$, $H_\phi(\eta) = \inf_{\alpha \in \mathbb{R}} C_\phi^\eta(\alpha)$, $H_\phi^-(\eta) = \inf_{\alpha:\alpha(2\eta-1)\leq 0} C_\phi^\eta(\alpha)$. As in [2], we require $\phi$ to be classification-calibrated, i.e., for any $\eta \neq 1/2$, $H_\phi^-(\eta) > H_\phi(\eta)$. We then define $\psi$ by $\psi = \tilde{\psi}^{**}$ where $\tilde{\psi}(\gamma) = H_\phi^-\left(\frac{1+\gamma}{2}\right) - H_\phi\left(\frac{1+\gamma}{2}\right)$ and $g^{**}$ is the Fenchel–Legendre biconjugate of $g$. We show that, by without loss of generality assuming that $\phi(0) = 1$ and $\inf_{\alpha \in \mathbb{R}} \phi(\alpha) = 0$, we can obtain a risk bound for $h_k$ that is the same as that for the single decision model optimization [2], as given in the following theorem.

**Theorem 1.** *For any classification-calibrated surrogate function* $\phi$ *satisfying* $\phi(0) = 1$ *and* $\inf_{\alpha \in \mathbb{R}} \phi(\alpha) = 0$, *any measurable function* $h_k$ *for predicting* $Y_k$, *we have*

$$\psi(R(h_k) - R^*) \leq R_\phi(h_k) - R_\phi^*,$$

*where* $\psi(\delta)$ *is a non-decreasing function mapping from* $[0, 1]$ *to* $[0, \infty)$.

*Proof Sketch.* The general idea is to factorize $R(h_k)$ and $R_\phi(h_k)$ into multiplications of $C_\phi^\eta(h_i(\mathbf{z}_i))$ and $C_\phi^\eta(h_i(\mathbf{z}_i))$ respectively, construct a lower bound of $R_\phi(h_k)$ as multiplications of $H_\phi^-(\eta)$, express $R^*$ and $R_\phi^*$ as multiplications of $H(\eta)$ and $H_\phi(\eta)$ respectively, and adopt properties of $H_\phi$, $H_\phi^-$, $\psi$ to derive the above inequality. For detailed proof, please refer to the supplementary file. □

The meaning of Theorem 1 clearly gives the following corollary.

**Corollary 1.** $R_\phi(h_k) \to R_\phi^*$ *indicates* $R(h_k) \to R^*$.

## 4 Experiments

**Experiment Setup.** We evaluate our method using both synthetic and real-world data. Table 1 provides a summary of two datasets' statistics. For the synthetic data, we manually define a causal

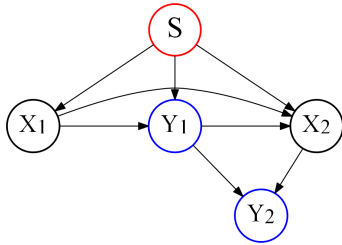

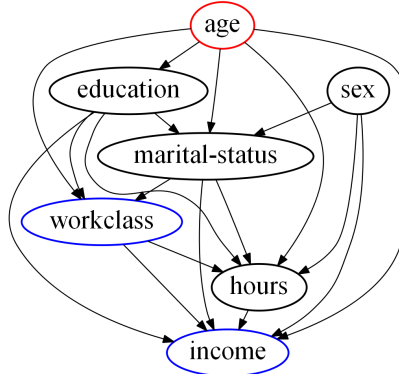

Figure 2: The causal graph for the synthetic dataset.

Figure 3: The causal graph for the Adult dataset.

graph with five variables $S, X_1, X_2, Y_1, Y_2$ shown in Fig. 2. Then, a conditional probability table is defined for each attribute over its parents, and the data is generated by sampling each attribute in topological order according to the conditional probability. For the real-world data, we use the Adult dataset [19] and build the causal graph by using the PC algorithm implemented in the Tetrad [25]. We follow the settings in [28] to select 7 out of 11 attributes and binarize their domain values. The significant threshold for conditional independence testing is set as 0.01, and three tiers in the partial order are used. We handle this imbalanced data using the over-sampling technique [18]. The resultant dataset consists of 10,1472 records. The causal graph is shown in Fig. 3. We treat `Age` as the protected attribute $S$, and `Workclass` and `Income` as two decisions $Y_1, Y_2$. By default, we use 0.05 as the threshold for judging fairness.

Table 1: Dataset statistics

| Dataset | #Instances | #Attributes | Sensitive Variable | Decision Variable |
|---------|-----------|-------------|--------------------|--------------------|
| Sythetic | 10,000 | 5 | $S$ | $Y_1, Y_2$ |
| Adult | 101,472 | 7 | age | workclass, income |

We design an evaluation process which simulates the real model deployment procedure. The dataset is randomly split to training and testing datasets. We deploy and evaluate the learned classifiers sequentially according to their topological order. The first classifier $h_1$ is deployed first, and evaluated on the original testing dataset. After that, it produces predicted decisions for $Y_1$, which are then used to re-generate the values of the subsequent variables in the order, as well as the true values of the next classifier $h_2$, by using the causal graph. In the end, we evaluate $h_2$ based on the re-generated data.

For training, our method (referred to as the joint method) formulates the optimization problem on the training data to learn all classifiers simultaneously. We also consider a simplified version of our method (referred to as the serial method) that learns classifiers sequentially following the topological order similarly to the deployment procedure. Each classifier only uses the direct parents of the label. After each classifier is learned, it is treated as a soft intervention such that the post-intervention distribution is inferred and used to train subsequent classifiers. We compare our methods with a baseline method (referred to the separate method) where each classifier is learned using the direct parents separately on the training data.

**Implementation.** All classifiers are implemented as empirical risk minimization classifiers where the logistic surrogate function is used. For unconstrained, separate, and serial methods, each classifier is learned individually as a convex optimization problem. Thus, we use the CVXPY package [7] to directly solve the unconstrained/constrained convex optimization problem. For the joint method, since the objective function and constraints are non-convex, we add constraints as penalty terms to the objective function and adopt PyTorch [22] to optimize it using the Adam optimizer. The convergence of Adam algorithms for non-convex optimization has been studied, e.g., in [5]. All experiments are conducted in a PC with 8GB RAM and Intel Core i5-1035G1 CPU.

Table 2: Accuracy and unfairness from Unconstrained, Separate, Serial and Joint methods on synthetic and Adult data (bold values indicate violation of fairness).

| Phase | | | Synthetic | | | | Adult | | | |
|---|---|---|---|---|---|---|---|---|---|---|
| | | | Uncons. | Separate | Serial | Joint | Uncons. | Separate | Serial | Joint |
| Train | $h_1$ | Acc. (%) | 80.32 | 75.35 | 75.35 | 75.35 | 55.71 | 55.64 | 55.63 | 55.63 |
| | | Unfairness | **0.15** | 0.01 | 0.01 | 0.01 | **0.15** | 0.05 | 0.05 | 0.05 |
| | $h_2$ | Acc. (%) | 90.13 | 75.79 | 84.02 | 82.77 | 76.75 | 71.17 | 68.90 | 69.31 |
| | | Unfairness | **0.23** | 0.04 | 0.03 | 0.04 | **0.24** | 0.10 | 0.10 | 0.10 |
| Test | $h_1$ | Acc. (%) | 80.70 | 75.54 | 75.54 | 75.54 | 55.63 | 55.56 | 55.57 | 55.57 |
| | | Unfairness | **0.15** | 0.01 | 0.01 | 0.01 | **0.15** | 0.05 | 0.05 | 0.05 |
| | $h_2$ | Acc. (%) | 89.95 | 77.06 | 84.16 | 82.09 | 77.07 | 73.33 | 68.91 | 69.40 |
| | | Unfairness | **0.13** | **0.09** | 0.03 | 0.03 | **0.23** | **0.17** | 0.10 | 0.10 |

**Results.** As discussed, since separate training does not consider the change in the data distribution caused by the deployment of new classifiers, it fails to achieve fairness in testing even if the classifier is fair in training. To demonstrate this, Table 2 shows the results of one typical setting for each method on both synthetic and Adult datasets, obtained from 5-fold cross-validation. For all methods, we manage to build classifiers that are fair in training[3]. We can see that, in testing, the serial and joint methods achieve consistent performance, but the separate method cannot guarantee to achieve fairness for $h_2$. We also did a grid search on thresholds $\tau_1, \tau_2$ on the synthetic data to find classifier pairs $h_1, h_2$ whose fairness is between -0.05 and 0.05 in training. Then, we evaluated these classifiers in testing. We observe that, even if we use the training data for testing to avoid any generalization error, in $71.43\%$ of these pairs produced by the separate method, $h_2$ exceeded the interval [-0.05, 0.05] and hence violated the fairness criterion. On the contrary, all classifiers produced by the serial and joint methods are fair in testing.

Comparing the serial and joint methods, they obtain similar results. This is expected since both of them apply the soft intervention to capture the model deployment. The advantage of the joint method is that it can adjust all classifiers simultaneously to obtain a better overall performance. This is not shown in current experiments probably due to the small scale of the problem. We will study whether the joint method would outperform the serial method in larger problems in our future work.

## 5    Conclusions and Future Work

In this paper, we proposed an approach that learns multiple fair classifiers from a static training dataset. We treated the deployment of each classifier as a soft intervention and inferred the distributions after the deployment as post-intervention distributions. We adopted surrogate functions to smooth the loss function and fair constraints to formulate the fair classification problem as a constrained optimization problem. In addition, we theoretically showed that combining multiple decision models in the optimization would not introduce additional surrogate errors. Experiments using both synthetic and real-world datasets show the advantage of our approach over the separate training method.

In our paper we assume the Markovian model. When the Markovian assumption is not satisfied, the causal model is called the semi-Markovian model, which faces the identifiability issue, i.e., causal effects may not be uniquely identified from the observational data. Most recently, the authors in [6] introduced $\sigma$-calculus to identify causal effects of soft interventions systematically. Extending our approach to identifiable situations in semi-Markovian models will be our further work.

**Reproducibility**. The source code and datasets are available at `https://github.com/yaoweihu/Fair-Multiple-Decision-Making`.

## Broader Impact

Our research could benefit any organization or system that uses computer algorithms to make important decisions, especially for large systems that consist of multiple decision tasks. By adopting our method, decision makers can build multiple decision models simultaneously just from one historical dataset and ensure that all decision models will be fair after the deployment. Our research could also benefit users who get involved in the system, in particular the users from disadvantage groups, by preventing them from receiving biased decisions.

## Acknowledgments and Disclosure of Funding

This work was supported in part by NSF 1646654, 1920920, and 1946391.

## Footnotes

[1]We use $Y_k$ to denote both decision label and predicted decision, and use soft intervention to distinguish between them: if the distribution is pre-interventional such as $P(y_k^+|\mathbf{z}_k)$, $Y_k$ is the label; if the distribution is post-interventional such as $P(y_k^+|do(s^+, h_1, \cdots, h_l))$, $Y_k$ is the prediction.

[2]For the sake of simple representation, we assume that $S$ has no parent in the causal graph.

[3]For the Adult dataset, we use 0.1 as the fairness threshold for $h_2$.

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
