[Supplementary Material]

# Fair Multiple Decision Making Through Soft Interventions (Supplementary File)

**Yaowei Hu**
University of Arkansas
yaoweihu@uark.edu

**Yongkai Wu**
Clemson University
yongkaw@clemson.edu

**Lu Zhang**
University of Arkansas
lz006@uark.edu

**Xintao Wu**
University of Arkansas
xintaowu@uark.edu

## 1 Proof of Theorem 1

**Theorem 1.** For any classification-calibrated loss function $\phi$ satisfying $\phi(0) = 1$ and $\inf_{\alpha \in \mathbb{R}} \phi(\alpha) = 0$, any measurable function $h_k$ for predicting $Y_k$, we have

$$\psi(R(h_k) - R^*) \leq R_\phi(h_k) - R_\phi^*,$$

where $\psi(\delta)$ is a non-decreasing function mapping from $[0, 1]$ to $[0, \infty)$.

**Lemma 1.** *For $\psi$, $H_\phi$ and $H_\phi^-$, they have following properties.*

1. *For $\lambda \in [0, 1]$ and $\gamma \in \mathbb{R}$, $\psi(\lambda\gamma) \leq \lambda\psi(\gamma)$.*

2. *$H_\phi^-(\eta) \geq H_\phi(\eta)$ for $\eta \in [0, 1]$.*

3. *$\eta \leq H_\phi(\eta)$ for $\eta \in [0, 1/2]$.*

4. *$\eta \leq 1 \leq H_\phi^-(\eta)$ for $\eta \in [0, 1]$.*

*Proof.* Parts 1,2,3 are proved in [1]. For Part 4, note that $H_\phi$ is concave and symmetric about $1/2$, meaning that it gets its minimum at $\eta = 0, 1$ and maximum at $\eta = 1/2$ [1]. We have $H_\phi(0) = H_\phi(1) = \inf_{\alpha \in \mathbb{R}} \phi(\alpha) = 0$. Meanwhile, we have $H_\phi(1/2) = 1/2 \cdot \inf_{\alpha \in \mathbb{R}}(\phi(\alpha) + \phi(-\alpha))$. Due to the convexity and symmetry between $\phi(\alpha)$ and $\phi(-\alpha)$, we can see that $H_\phi(1/2) = \phi(0) = 1$. Then, since $H_\phi$ is concave, we have $\eta H_\phi(1/2) + \bar{\eta} H_\phi(0) \leq H_\phi(\eta/2 + \bar{\eta} \cdot 0)$, which leads to $\eta \leq H_\phi(\eta/2) \leq H_\phi(\eta)$ for $\eta \in [0, 1/2]$.

For Part 5, note that $H_\phi^-$ is concave on $[0, 1/2]$ and on $[1/2, 1]$ and also symmetric about $1/2$ [1]. Since $H_\phi^-(1/2) = H_\phi(1/2) = 1$ and $H_\phi^-(0) = H_\phi^-(1) = \inf_{\alpha \leq 0} \phi(\alpha) = \phi(0) = 1$, we have $H_\phi^-(\eta) \geq 1 \geq \eta$. $\qquad\square$

Next, we first prove Theorem 1 based on the toy example in the main paper, and then explain how this proof can be extended to general situations.

### 1.1 Proof of Theorem 1 Based on Toy Example

*Proof of Theorem 1.* The causal graph of the toy example is shown in Fig. 1. In the example, we have two classifiers $h_1, h_2$. Note that $R_\phi(h_1)$ is the same as that of a single decision model, so we

Figure 1: The toy model.

focus on $R_\phi(h_2)$. Denoting $\mathbf{Z} = \{S, X_1, X_2\}$, we define

$$c_1(\mathbf{z}) = \frac{P(y_1^+|s, x_1, x_2)}{P(y_1^+|s, x_1)} + \frac{P(y_1^-|s, x_1, x_2)}{P(y_1^-|s, x_1)},$$

and define

$$\eta_1(\mathbf{z}) = \frac{P(y_1^-|s, x_1, x_2)}{c_1(\mathbf{z})}, \quad \eta_2(\mathbf{z}) = P(y_2^+|x_2),$$

and

$$\bar{\eta}_1(\mathbf{z}) = 1 - \eta_1(\mathbf{z}), \quad \bar{\eta}_2(\mathbf{z}) = 1 - \eta_2(\mathbf{z}),$$

For simplifying representation, in the remaining of this file we omit $(\mathbf{z})$ in all expressions.

Note that

$$R_\phi(h_2) = \mathbb{E}_{\mathbf{z}} \left[ c_1 \left( \eta_1 \eta_2 \phi(h_1(x_1))\phi(h_2(x_2)) + \bar{\eta}_1 \eta_2 \phi(-h_1(x_1))\phi(h_2(x_2)) \right. \right.$$

$$\left. + \eta_1 \bar{\eta}_2 \phi(h_1(x_1))\phi(-h_2(x_2)) + \bar{\eta}_1 \bar{\eta}_2 \phi(-h_1(x_1))\phi(-h_2(x_2))) \right]$$

$$= \mathbb{E}_{\mathbf{z}} \left[ c_1(\eta_1 \phi(h_1(x_1)) + \bar{\eta}_1 \phi(-h_1(x_1) > 0))(\eta_2 \phi(h_2(x_2)) + \bar{\eta}_2 \phi(-h_2(x_2))) \right]$$

$$= \mathbb{E}_{\mathbf{z}} \left[ c_1 C_\phi^{\eta_1}(h_1(x_1)) C_\phi^{\eta_2}(h_2(x_2)) \right],$$

we can express $R_\phi(h_2)$ using the generic $\phi$-conditional risk $C_\phi^\eta(\alpha)$. According to the definition of $R_\phi^*$, we correspondingly have

$$R_\phi^* = \mathbb{E}_{\mathbf{z}} \left[ c_1 H_\phi(\eta_1) H_\phi(\eta_2) \right].$$

Similarly we can also express $R(h_2)$ and $R^*$ as

$$R(h_2) = \mathbb{E}_{\mathbf{z}} \left[ c_1 C^{\eta_1}(h_1(x_1)) C^{\eta_2}(h_2(x_2)) \right],$$

$$R^* = \mathbb{E}_{\mathbf{z}} [c_1 H(\eta_1) H(\eta_2)],$$

where $C^\eta(\alpha)$ and $H(\eta)$ are defined by replacing $\phi$ with $\mathbb{1}$ in $C_\phi^\eta(\alpha)$ and $H_\phi(\eta)$. Note that $H(\eta)$ is always obtained when the sign of $\alpha$ is the same as the sign of $\eta - 1/2$.

Denote by $\alpha^*$ the signs of solutions $\{\text{sign}(\eta_1 - 1/2), \text{sign}(\eta_2 - 1/2)\}$. Then, we have

$$R(h_2) - R^* = \mathbb{E}_{\mathbf{z}} \left[ c_1 \left( C^{\eta_1}(h_1(x_1)) C^{\eta_2}(h_2(x_2)) - H(\eta_1) H(\eta_2) \right) \right]$$

$$= \mathbb{E}_{\mathbf{z}} \left[ c_1 \mathbb{1}(\text{sign}(h) \neq \alpha^*) \left( C^{\eta_1}(h_1(x_1)) C^{\eta_2}(h_2(x_2)) - H(\eta_1) H(\eta_2) \right) \right].$$

Since $\psi$ is convex [1], it follows that

$$\psi(R(h_2) - R^*) \leq \mathbb{E}_{\mathbf{z}} \left[ c_1 \mathbb{1}(\text{sign}(h) \neq \alpha^*) \psi \left( C^{\eta_1}(h_1(x_1)) C^{\eta_2}(h_2(x_2)) - H(\eta_1) H(\eta_2) \right) \right].$$

Without loss of generality, assume $\eta_1 \leq \bar{\eta}_1$ and $\eta_2 \leq \bar{\eta}_2$. Thus, according to the definition, $H(\eta_1) = \eta_1$ and $H(\eta_2) = \eta_2$. Then, we want to show that for any $h_1, h_2$ whose signs are not equivalent to $\alpha^*$, we have

$$\psi \left( C^{\eta_1}(h_1(x_1)) C^{\eta_2}(h_2(x_2)) - H(\eta_1) H(\eta_2) \right) \leq H_\phi^-(\eta_1) H_\phi^-(\eta_2) - H_\phi(\eta_1) H_\phi(\eta_2). \quad (1)$$

To this end, we consider two cases: (1) only one classifier from $h_1, h_2$ makes the prediction that is opposite to $\alpha^*$; and (2) both $h_1, h_2$ make predictions that are opposite to $\alpha^*$.

For Case (1), assume that $h_1$ makes the opposite prediction. Thus, $C^{\eta_1}(h_1(x_1)) = \bar{\eta}_1$, and $C^{\eta_2}(h_2(x_2)) = \eta_2$. Then, we have

$$\psi\left(C^{\eta_1}(h_1(x_1))C^{\eta_2}(h_2(x_2)) - H(\eta_1)H(\eta_2)\right) = \psi\left((\bar{\eta}_1 - \eta_1)\eta_2\right).$$

Based on Lemma 1, Part 1, it follows that

$$\psi\left((\bar{\eta}_1 - \eta_1)\eta_2\right) \leq \eta_2 \psi\left(\bar{\eta}_1 - \eta_1\right) = \eta_2\left(H_\phi^-(\eta_1) - H_\phi(\eta_1)\right).$$

Based on Lemma 1, Part 3, we have $\eta_2 \leq H_\phi(\eta_2)$. So it follows that

$$\psi\left((\bar{\eta}_1 - \eta_1)\eta_2\right) \leq \left(H_\phi^-(\eta_1) - H_\phi(\eta_1)\right)H_\phi(\eta_2).$$

Based on Lemma 1, Part 2, we prove Eq. (1).

For Case (2), we have $C^{\eta_1}(h_1(x_1)) = \bar{\eta}_1$, and $C^{\eta_2}(h_2(x_2)) = \bar{\eta}_2$. Thus,

$$\psi\left(C^{\eta_1}(h_1(x_1))C^{\eta_2}(h_2(x_2)) - H(\eta_1)H(\eta_2)\right) = \psi\left(\bar{\eta}_1\bar{\eta}_2 - \eta_1\eta_2\right).$$

Without loss of generality, assume $\eta_1 \leq \eta_2$, i.e., $\bar{\eta}_2 \leq \bar{\eta}_1$. We have that

$$
\begin{aligned}
\bar{\eta}_1\bar{\eta}_2 - \eta_1\eta_2 &= \bar{\eta}_1\bar{\eta}_2 - \eta_1\eta_2 - \eta_1\bar{\eta}_2 + \eta_1\bar{\eta}_2 \\
&= \bar{\eta}_2(\bar{\eta}_1 - \eta_1) + \eta_1(\bar{\eta}_2 - \eta_2) \\
&\leq \bar{\eta}_1(\bar{\eta}_1 - \eta_1) + \eta_1(\bar{\eta}_2 - \eta_2).
\end{aligned}
\tag{2}
$$

Since $\psi$ is convex, we have

$$
\begin{aligned}
\psi\left(\bar{\eta}_1\bar{\eta}_2 - \eta_1\eta_2\right) &\leq \psi\left(\bar{\eta}_1(\bar{\eta}_1 - \eta_1) + \eta_1(\bar{\eta}_2 - \eta_2)\right) \\
&\leq \bar{\eta}_1\psi(\bar{\eta}_1 - \eta_1) + \eta_1\psi(\bar{\eta}_2 - \eta_2).
\end{aligned}
$$

According to the definition of $\psi$, we have $\psi(\bar{\eta} - \eta) = H_\phi^-(\eta) - H_\phi(\eta)$. According to Lemma 1, Parts 4&3, we have $\bar{\eta}_1 \leq 1 \leq H_\phi^-(\eta_2)$, $\eta_1 \leq H_\phi(\eta_1)$. As a result, we have

$$\psi\left(\bar{\eta}_1\bar{\eta}_2 - \eta_1\eta_2\right) \leq H_\phi^-(\eta_2)\left(H_\phi^-(\eta_1) - H_\phi(\eta_1)\right) + H_\phi(\eta_1)\left(H_\phi^-(\eta_2) - H_\phi(\eta_2)\right)$$

which proves Eq. (1).

Finally, we have

$$
\begin{aligned}
\psi\left(R(h_2) - R^*\right) &\leq \mathbb{E}_{\mathbf{z}}\left[c_1\mathbb{1}(\text{sign}(h) \neq \alpha^*)\left(H_\phi^-(\eta_1)H_\phi^-(\eta_2) - H_\phi(\eta_1)H_\phi(\eta_2)\right)\right] \\
&\leq \mathbb{E}_{\mathbf{z}}\left[c_1\mathbb{1}(\text{sign}(h) \neq \alpha^*)\left(C_\phi^{\eta_1}(h_1(x_1))C_\phi^{\eta_2}(h_2(x_2)) - H_\phi(\eta_1)H_\phi(\eta_2)\right)\right] \\
&\leq \mathbb{E}_{\mathbf{z}}\left[c_1\left(C_\phi^{\eta_1}(h_1(x_1))C_\phi^{\eta_2}(h_2(x_2)) - H_\phi(\eta_1)H_\phi(\eta_2)\right)\right] \\
&= R_\phi(h_2) - R_\phi^*.
\end{aligned}
$$

$\square$

## 1.2 Extending to General Situations

We prove that Theorem 1 can be extended to $h_3$, then, it can be similarly extended to any $k$. Note that the key is to prove

$$
\begin{aligned}
&\psi\left(C^{\eta_1}(h_1(x_1))C^{\eta_2}(h_2(x_2))C^{\eta_3}(h_3(x_3)) - H(\eta_1)H(\eta_2)H(\eta_3)\right) \\
&\leq H_\phi^-(\eta_1)H_\phi^-(\eta_2)H_\phi^-(\eta_3) - H_\phi(\eta_1)H_\phi(\eta_2)H_\phi(\eta_3).
\end{aligned}
\tag{3}
$$

Similarly, we consider three cases: (1) only one classifier from $h_1, h_2, h_3$ makes the prediction that is opposite to $\alpha^*$; (2) two classifiers from $h_1, h_2, h_3$ make predictions that are opposite to $\alpha^*$; and (3) all three classifiers make predictions that are opposite to $\alpha^*$.

For Case (1), the proof is similar to that in Section 1.1.

For Case (2), assume that $h_1, h_2$ make the opposite predictions. Then, we have

$$\psi\left(C^{\eta_1}(h_1(x_1))C^{\eta_2}(h_2(x_2))C^{\eta_3}(h_3(x_3)) - H(\eta_1)H(\eta_2)H(\eta_3)\right)$$
$$= \psi\left((\bar{\eta}_1\bar{\eta}_2 - \eta_1\eta_2)\eta_3\right) \le \eta_3\psi\left(\bar{\eta}_1\bar{\eta}_2 - \eta_1\eta_2\right).$$

Thus, based on Eq. (1), we can prove Eq. (3).

For Case (3), without loss of generality, assume that $\eta_1 \le \eta_2 \le \eta_3$, i.e., $\bar{\eta}_3 \le \bar{\eta}_2 \le \bar{\eta}_1$. Then, we have

$$\psi\left(C^{\eta_1}(h_1(x_1))C^{\eta_2}(h_2(x_2))C^{\eta_3}(h_3(x_3)) - H(\eta_1)H(\eta_2)H(\eta_3)\right)$$
$$= \psi\left(\bar{\eta}_1\bar{\eta}_2\bar{\eta}_3 - \eta_1\eta_2\eta_3\right)$$
$$= \psi\left(\bar{\eta}_3(\bar{\eta}_1\bar{\eta}_2 - \eta_1\eta_2) + \eta_1\eta_2(\bar{\eta}_3 - \eta_3)\right).$$

Based on Eq. (2), it follows that

$$\psi\left(\bar{\eta}_3(\bar{\eta}_1\bar{\eta}_2 - \eta_1\eta_2) + \eta_1\eta_2(\bar{\eta}_3 - \eta_3)\right)$$
$$= \psi\left(\bar{\eta}_3\bar{\eta}_2(\bar{\eta}_1 - \eta_1) + \bar{\eta}_3\eta_1(\bar{\eta}_2 - \eta_2) + \eta_1\eta_2(\bar{\eta}_3 - \eta_3)\right)$$
$$\le \psi\left(\bar{\eta}_1\bar{\eta}_2(\bar{\eta}_1 - \eta_1) + \bar{\eta}_2\eta_1(\bar{\eta}_2 - \eta_2) + \eta_1\eta_2(\bar{\eta}_3 - \eta_3)\right)$$
$$\le \bar{\eta}_1\bar{\eta}_2\psi\left(\bar{\eta}_1 - \eta_1\right) + \bar{\eta}_2\eta_1\psi\left(\bar{\eta}_2 - \eta_2\right) + \eta_1\eta_2\psi\left(\bar{\eta}_3 - \eta_3\right).$$

Then, since $\bar{\eta}_1 \le 1 \le H_\phi^-(\eta_2), \bar{\eta}_2 \le 1 \le H_\phi^-(\eta_3), \eta_1 \le H_\phi(\eta_1), \eta_2 \le H_\phi(\eta_2)$, it follows that

$$\bar{\eta}_1\bar{\eta}_2\psi\left(\bar{\eta}_1 - \eta_1\right) + \bar{\eta}_2\eta_1\psi\left(\bar{\eta}_2 - \eta_2\right) + \eta_1\eta_2\psi\left(\bar{\eta}_3 - \eta_3\right)$$
$$\le H_\phi^-(\eta_2)H_\phi^-(\eta_3)\left(H_\phi^-(\eta_1) - H_\phi(\eta_1)\right) + H_\phi^-(\eta_3)H_\phi(\eta_1)\left(H_\phi^-(\eta_2) - H_\phi(\eta_2)\right)$$
$$\quad + H_\phi(\eta_1)H_\phi(\eta_2)\left(H_\phi^-(\eta_3) - H_\phi(\eta_3)\right)$$
$$= H_\phi^-(\eta_1)H_\phi^-(\eta_2)H_\phi^-(\eta_3) - H_\phi(\eta_1)H_\phi(\eta_2)H_\phi(\eta_3),$$

which proves the Eq. (3).

## References

[1] Peter L Bartlett, Michael I Jordan, and Jon D McAuliffe. Convexity, classification, and risk bounds. *Journal of the American Statistical Association*, 101(473):138–156, 2006.