[Reviews · NeurIPS 2020]

Review 1

Summary and Contributions: The authors propose an approach to learn multiple fair classifiers by treating each one as a soft intervention and infer their distribution.

Strengths: The paper studies a very interesting problem that fairness is not guaranteed under composition. There are not many existing work on this topic, however this is an important and challenging topic which requires attention and I very much encourage the authors to continue working on it. Their approach is based on SCM that can address current causal fairness definitions such as counterfactual fairness, which make their approach not specific to a single fairness notion.

Weaknesses: It is not necessarily a weakness, but it is interesting to see how they approach can extend to online setup that training data is changing during time. Their code is not available as a supplementary material and it would be useful for the community if the authors could make their code publicly available. I'd suggest the authors to narrow down their claim and only mention their contribution in which they consider the problem based on SCM and how they formulate it as a constrained optimization problem.

Correctness: The authors provide details of deriving the loss function and constraints and formulate the problem clearly.

Clarity: The paper is well-written and easy to read.

Relation to Prior Work: The authors claim that this work is the first work that studies multiple fair classifiers. However, there exists works that study fairness under composition. Besides multiple work by Cyntia Dwork, there are some recent attempts to study this problem such as: Wang, Xuezhi, et al. "Practical Compositional Fairness: Understanding Fairness in Multi-Task ML Systems." arXiv preprint arXiv:1911.01916 (2019). Hence, as I mentioned above, I'd suggest the authors to narrow down their claim.

Reproducibility: No

Additional Feedback: - Can authors say something about scalability of their approach? - Can the authors make their code publicly available? - What is the capacity and complexity of their model when there are multiple interventions involved in calculating one variable? - What is the bottleneck when there are more than one sensitive attribute? ================================= EDIT AFTER AUTHOR RESPONSE ================================= I am satisfied with the authors' response. I am willing to raise my score to 7.


Review 2

Summary and Contributions: This paper discussed the problem of learning multiple fair classifiers from a static training dataset. The main contributions of this paper lie in the following three aspects: 1. Instead of building decision models separately without taking upstream models into consideration, the proposed method can fairly learn multiple classifiers by introducing fairness constrains. 2. Applying the soft intervention to formulate the loss function as well as the fairness constraints. 3. Provide the excess risk bound for the final loss function. The experimental results on the synthetic and real dataset demonstrated the advantage of the proposed method to some extent.

Strengths: 1. A novel framework to combine multiple classifiers is proposed, which can reduce the risk of unfairness in decision making. 2. The toy example is well explained, which helps the audience to understand the core idea of fair classification. 3. The method does show an improvement over some other comparing algorithms in this area. 4. The mathematical formulation is strong, and the mathematical derivation are reasonable (I do not check all of them).

Weaknesses: 1. The proposed method is based on a pre-defined causal graph, which has limitations if the causal graph is unavailable. In the experimental results sections, the authors only showed the results with the graph constructed by the PC algorithm. It is not clear how the way of graph construction affects the final results. 2. The optimization details for the objective function is missed. 3. This paper lacks the complexity analysis for the proposed method. 4. Only validating the proposed method on one real dataset (i.e., Adult) cannot guarantee its applicability in the wide spectrum of real-world applications. 5. The assumption about the mutual independency of the exogenous variables are too strong to be satisfied in real-world applications.

Correctness: The methodology part is generally correct with reasonable mathematical derivation.

Clarity: This paper is presented in clear English and easy for readers to follow.

Relation to Prior Work: Yes, they clearly clarified the difference between current study and previous works.

Reproducibility: Yes

Additional Feedback: N/A.


Review 3

Summary and Contributions: This paper considers fairness in a sequential setting, where the decisions from upstream models feed into downstream models. The paper uses a causal inference framework to model the interaction between upstream and downstream models. They aim to enforce fairness jointly for multiple classifiers in the pipeline simultaneously, which they formulate as a constrained optimization problem (with standard surrogate loss functions to replace the 0/1 loss). Their framework supports any fairness notions based on structural causal models (SCMs), including “total effect” and “counterfactual fairness.” Their constrained optimization problem involves multiplying the objective by a surrogate loss function (e.g. hinge loss) for each classifier in the pipeline. As a theoretical contribution, they show that the compounded multiplication of these surrogate loss functions doesn’t hurt convergence: Theorem 1 says that convergence with surrogate loss functions implies convergence with the original 0/1 loss. As an empirical contribution, they compare their method of jointly enforcing fairness across multiple models to a baseline where each model is trained separately. They use a simulation and a benchmark dataset with a learned casual graph.

Strengths: - Theorem 1 seems to be a nontrivial and useful result. The study of how the compounded multiplication of surrogate loss functions affects convergence is important when surrogate loss functions are applied in practice. - The experimental comparisons seem interesting and valuable. The comparison to separately trained models seems like a good baseline. The comparison between serial and jointly trained methods is also useful, even though their results didn’t uncover any significant differences between these two methods. - The paper does a good job providing the relevant background in Section 2 on causal inference necessary for understanding their approach. - The problem of enforcing fairness for multiple models feeding into each other simultaneously seems practically useful.

Weaknesses: - As the authors mentioned, their experimental results don’t effectively show a difference between the serial and joint methods. The authors mention that this could be due to the small scale of the experiments. I agree that including an experiment with a larger dataset would make this experimental section stronger. - There are issues with clarity in the notation and technical results (see below comments on “Clarity”). - Can the authors provide more details on the “separate” baseline in the experiments? Currently, Line 292 just says, “each classifier is learned separately on the training data.” However, can the authors specify which features each separate classifier uses? Does each classifier only use the direct parents of each label in the causal graph, or does each classifier use all available features? The same question applies to the Serial comparison. I could not find this information in the attached supplemental pdf. - The empirical results reported from Lines 300-304 are not at all clear to me. I’m not sure what point is being made here, what takeaways I should have, or what the 71.43% number really means. Can the authors describe these results more clearly, perhaps by further describing what they mean when they say multiple classifiers are “produced by” a given method? - The authors briefly mention that they perform “oversampling” on the Adult dataset to handle imbalanced data. Somehow, they obtain a dataset with 101,472 examples, when the original Adult dataset contains 48842 examples. However, they do not explicitly define what they mean by “imbalanced,” nor do they specifically outline what oversampling procedure they use (other than referencing a toolbox by Lemaitre et al.). Can the authors more explicitly outline their oversampling procedure?

Correctness: Their substitution of surrogate loss functions in the constrained optimization problem seems reasonable. I haven’t carefully checked the proof of Theorem 1.

Clarity: The notation and technical results could use some rewriting and reframing for clarity. - Is there a typo in Definition 1? Definition 1 says “for each predicted decision \hat{Y}_k...”, but \hat{Y}_k is nowhere to be found in the expression below. Do they mean to replace \hat{Y}_k with Y_k? - In general, the introduction of the notation with \hat{Y}_k is confusing. I don’t see \hat{Y} or \hat{y}_k ever referenced again after being defined on Line 138. - Line 211-212 is also confusingly written. Do they mean, “y_i = y^+ if h_i(z_i) \leq 0 and y_i = y^- if h_i(z_i) < 0”? - The paper uses Y_k to represent both a classification label (equation 3) and a prediction from a previous classifier (equation 5). This shift in representation of Y_k from label to prediction is confusing, and it would be helpful for the authors to clarify when this shift has occurred, or to use a different variable to represent the predictions. (Perhaps the authors could actually use the \hat{Y} variable that they defined for predictions?) - The expressions on page 5 are difficult to parse, particularly on Line 204 and Line 207. It would be helpful for the authors to break down these expressions into smaller parts, perhaps using placeholder functions for each term. Or, perhaps the authors could skip directly to the final formulations of R_phi(h_k) and T_phi(h_k), and move the previous derivations on lines 220, 221, and 226 to a proof. - Lines 253-262 are dense and hard to parse. Perhaps the authors could more succinctly summarize the general ideas of the proof (as done in the proof sketch on Line 268), and leave these specific technical details in Lines 253-262 in the proof itself.

Relation to Prior Work: Their approach and formulation seems novel for enforcing causal inference fairness constraints in multiple models simultaneously. The related work section references prior work that studies how individuals drop out at various stages in a pipeline of classifiers, but this appears to be a different setup.

Reproducibility: Yes

Additional Feedback: Overall, my score is marginally below the acceptance threshold due to issues with clarity of the technical results of the paper. The general idea of enforcing causal inference fairness notions on multiple models simultaneously seems interesting, and Theorem 1 seems useful. If the authors are able to answer some of my questions on the experimental section and improve some of the clarity of the notation and presentation of technical results, then I would improve my score. ================================= EDIT AFTER AUTHOR RESPONSE ================================= The authors addressed my main clarity concerns in the author response. They clarified the notation, and clarified my questions on the experimental setup. I find these useful, and am willing to raise my score to 6.

[Author Response · NeurIPS 2020]

We thank the reviewers for the constructive comments. We will revise the paper accordingly.

**Reviewer#1**
- Our paper exactly deals with the dynamic situation where feature distribution could change over time due
to the deployment of classifiers. Regarding the situation where training data arrive sequentially, it is a different setup.

- **Our code is available online** as given in lines 88-89 in the supplementary file.

- As we pointed out in lines 76-79, Dwork's compound decision-making process or pipelines differ from our setting
in that individuals drop out at any stage and classification in subsequent stages depends on the remaining cohort of
individuals. Wang's paper assumes that multiple functions over the same set of attributes are multiplied to produce an
overall score. To the best of our knowledge, our paper is the first work to study the fair learning scenario where there
exist multiple related classifiers at different stages and the feature distribution may change due to the deployment of
classifiers. We will narrow down our claim to be more accurate.

- For complexity and scalability, if there is only one classifier, then our problem formulation is a convex constrained
optimization. If there are multiple classifiers, neither the loss function nor constraints are convex. However, their
gradients can be easily computed since each classifier is involved as a single term in the multiplication (e.g., Eqs. (7,8)).
Thus, adaptive gradient methods for non-convex optimization such as Adam can be straightforwardly adopted. The
convergence of Adam-type algorithms for non-convex optimization has been studied, e.g., in [Chen, et.al. ICLR'19].

- Multiple sensitive attributes are not a bottleneck of the paper as most fairness notions can be easily extended to handle
multiple sensitive attributes. For example, $P(y^+|do(s))$ in the total effect can be extended to $P(y^+|do(\mathbf{s}))$ where $\mathbf{s}$ is a
value assignment to the combination of multiple sensitive attributes.

**Reviewer#3**
- The assumption that the causal graph is given is common in the fairness research based on Pearl's struc-
tural causal models. In addition to the PC algorithm, there are also quite a number of algorithms to build causal graphs
from the data. The sensitivity of causal inference on the learned causal graph structure is beyond the scope of our paper.

- In optimization, we actually add constraints to the objective function as regularization terms. As mentioned in
responses to Reviewer#1, the gradients can be easily computed. Then, we adopt Adam for the optimization.

- Regarding complexity, please refer to the corresponding response to Reviewer#1.

- We plan to do experiments with more datasets.

- In this paper we assume Markovian models for simplicity. However, our method can also be extended to scenarios
where the Markovian assumption does not hold, a.k.a., semi-Markovian models. As discussed in lines 318-322, we will
explore the use of $\sigma$-calculus for judging identifiability and computing post-intervention distributions. Furthermore, in
the case of unidentifiable, we will resort to bounding approaches to deal with soft interventions.

**Reviewer#4**
- Clarity. In our paper, we use $y_k$ to denote both classification label and prediction, and use soft intervention
to distinguish between them: if the distribution is pre-interventional (i.e., observational), such as $P(y_k^+|\mathbf{z}_k)$, $y_k^+$ is the
label; if the distribution is post-interventional, such as $P(y_k^+|do(\cdots, h_k, \cdots))$ or $P_{h_k}(y_k^+|\mathbf{z}_k)$, $y_k^+$ is the prediction.
After we convert post-intervention distributions to observable distributions, all probabilities are to be estimated from the
training data. We originally planed to use $y_k$ to denote the label and $\hat{y}_k$ to denote the prediction. However, this would
make the notations too tedious and decrease the readability since most $y_k^+, y_k^-$ in all equations would become $\hat{y}_k^+, \hat{y}_k^-$.
On the other hand, from the viewpoint of soft intervention, the prediction is simply an interventional variant of the label
upon performing the soft intervention and hence can be distinguished by soft intervention without ambiguity. We will
remove notations $\hat{Y}_k$ and $\hat{y}_k$ and more clearly state our notations. Regarding your specific questions: (1) In Definition 1,
$y_k^+$ means the prediction. (2) In lines 211-212, $y_i$ also means the prediction. (3) We will also revise lines 204, 207, and
253-262 of the paper to improve the readability following your suggestions.

- For the separate method, each classifier uses the direct parents of each label and is learned directly from the training
data. For the serial method, each classifier also uses the direct parents of each label but is learned from the distribution
after upstream classifiers are deployed.

- For the separate method, we did a grid search on $\tau_1, \tau_2$ to find classifier pairs $h_1, h_2$ whose fairness is between -0.05
and 0.05 in training. Then, we evaluated these classifiers in testing and found that in $71.43\%$ of these pairs, $h_2$ exceeded
the interval [-0.05, 0.05] and hence violated the fairness criterion.

In the Adult dataset, `Workclass` and `Income` are two decisions with disproportionate (imbalanced) ratios (31:69 for
`Workclass` and 24:86 for `Income`). We oversampled the data twice to adjust the ratios of two decisions to 48:52 and
50:50 respectively which helps us focus on utility-fairness evaluation without distraction from imbalanced classifiers.

[Meta-Review · NeurIPS 2020]

All the reviewers agreed that the paper tackles an interesting problem and propose a reasonable solution. The rebuttal persuaded the reviewers to raise their scores and support acceptance. I encourage the authors to take into account the lengthy, informative reviews when preparing the final version of the paper.